# Interfacial magnetic spin Hall effect in van der Waals Fe₃GeTe₂/MoTe₂ heterostructure

Yudi Dai[1,9], Junlin Xiong [1,9], Yanfeng Ge [2,9], Bin Cheng [3 ✉], Lizheng Wang [1], Pengfei Wang [1], Zenglin Liu[1], Shengnan Yan[1], Cuiwei Zhang [4], Xianghan Xu [5], Youguo Shi[4], Sang-Wook Cheong [5], Cong Xiao[6,7,8 ✉], Shengyuan A. Yang [6], Shi-Jun Liang [1 ✉] & Feng Miao [1 ✉]

The spin Hall effect (SHE) allows efficient generation of spin polarization or spin current through charge current and plays a crucial role in the development of spintronics. While SHE typically occurs in non-magnetic materials and is time-reversal even, exploring time-reversal-odd (*T*-odd) SHE, which couples SHE to magnetization in ferromagnetic materials, offers a new charge-spin conversion mechanism with new functionalities. Here, we report the observation of giant *T*-odd SHE in Fe₃GeTe₂/MoTe₂ van der Waals heterostructure, representing a previously unidentified interfacial magnetic spin Hall effect (interfacial-MSHE). Through rigorous symmetry analysis and theoretical calculations, we attribute the interfacial-MSHE to a symmetry-breaking induced spin current dipole at the vdW interface. Furthermore, we show that this linear effect can be used for implementing multiply-accumulate operations and binary convolutional neural networks with cascaded multi-terminal devices. Our findings uncover an interfacial *T*-odd charge-spin conversion mechanism with promising potential for energy-efficient in-memory computing.

The development of high-density memory and computing has been a major focus in the field of spintronics[1–10]. Ferromagnets with perpendicular magnetic anisotropy (PMA) are considered as ideal material platforms for realizing spintronic devices[11–14], due to their scalability and compatibility with CMOS technology, and the ability to manipulate their magnetization through charge-spin conversion mechanisms[15,16] such as spin Hall effect[17–23], interfacial Edelstein effect[24,25]. Coupling the charge-spin conversion with the magnetization in ferromagnetic systems with PMA would provide a unique knob for reconfiguring the charge-spin conversion, introducing an additional pathway for developing novel spintronics devices. Recently, it has

been proposed that such magnetization-determined charge-spin conversion can be realized through time-reversal-odd SHE[26,27], in which the switching of magnetization will reverse the spin Hall current. However, magnetic materials hosting such *T*-odd SHE are still rare and only limited in antiferromagnetic systems, i.e., non-collinear bulk antiferromagnet Mn₃Sn[18,28] and collinear bulk antiferromagnet RuO₂[20,27,29]. The study of *T*-odd SHE in ferromagnetic systems remains unexplored so far.

Van der Waals (vdW) ferromagnetic heterostructures comprised of vdW materials with PMA and strong spin-orbit coupling (SOC) offer an unprecedented opportunity for exploring magnetization-related

¹National Laboratory of Solid State Microstructures, Institute of Brain-Inspired Intelligence, School of Physics, Collaborative Innovation Center of Advanced Microstructures, Nanjing University, Nanjing 210093, China. ²Research Laboratory for Quantum Materials, Singapore University of Technology and Design, Singapore, Singapore. ³Institute of Interdisciplinary Physical Sciences, School of Science, Nanjing University of Science and Technology, Nanjing 210094, China. ⁴Institute of Physics, Chinese Academy of Sciences, 100190 Beijing, China. ⁵Center for Quantum Materials Synthesis and Department of Physics and Astronomy, Rutgers, The State University of New Jersey, Piscataway, NJ 08854, USA. ⁶Institute of Applied Physics and Materials Engineering, University of Macau, Taipa, Macau, SAR, China. ⁷Department of Physics, University of Hong Kong, Hong Kong, China. ⁸HKU-UCAS Joint Institute of Theoretical and Computational Physics at Hong Kong, Hong Kong, China. ⁹These authors contributed equally: Yudi Dai, Junlin Xiong, Yanfeng Ge. ✉e-mail: bincheng@njust.edu.cn; congxiao@um.edu.mo; sjliang@nju.edu.cn; miao@nju.edu.cn

spin-charge conversion mechanisms. Without suffering the issues of lattice mismatch and interface disorders as in traditional ferromagnetic heterostructures[30], vdW interfaces are atomically sharp, free of dangling bonds and structural defects, providing more freedoms and better control in symmetry engineering[31–33]. Notably, such symmetry engineering at vdW heterointerface plays a vital role in generating unconventional charge-spin conversion mechanisms, such as non-orthogonal SHE for realizing field-free current-induced perpendicular magnetization reversal[34]. Moreover, strong interaction between the two constituting materials, despite the existence of a vdW gap region, can lead to coherent distribution of active electronic states across the vdW interface. In this way, vdW ferromagnetic heterostructures offer ideal platforms to explore many fascinating physics not available in the traditional bulk materials and are of paramount importance to the development of novel spin-based memory and computing technologies[35,36].

In this work, we present the discovery of a time-reversal-odd ($T$-odd) charge-spin conversion over the vdW interface of Fe₃GeTe₂/MoTe₂ heterostructure and demonstrate its promising application in in-memory computing devices. We observe that a charge current induces a pure spin current that is locked to the perpendicular magnetization of Fe₃GeTe₂. Our findings show a high efficiency of charge-spin conversion with a large spin Hall angle of 0.67. This giant interfacial-MSHE arises from the intricate interplay of symmetry

breaking and band geometric properties at the interface of Fe₃GeTe₂/MoTe₂ vdW heterostructure, as verified by our experiments and theoretical calculations. Based on this interfacial spin-charge-conversion, we propose a spintronic device called mem-transformer and prove its effectiveness in multiply-accumulate operations and artificial neural network. Our work paves the way for exploring a wealth of symmetry-breaking related physics and new computing technologies.

## Results

### SHE in Fe₃GeTe₂/MoTe₂ vdW heterostructure

We created a vdW interface by utilizing vdW layered-structure materials Fe₃GeTe₂ (FGT) and MoTe₂. The fabrication process is described in further detail in the Methods section. FGT is a ferromagnetic metal with strong perpendicular magnetic anisotropy[37] and exhibits a Curie temperature as high as 220 K in bulk form[38–42]. MoTe₂ as a non-magnetic metal is known for its high charge-spin conversion efficiency[43,44] and robust spin diffusion over several micrometers[44,45], making it suitable for efficient detection of spin current through inverse spin Hall effect (ISHE). Figure 1a schematically shows the multi-terminal vdW heterostructure device that we fabricated, which was used to conduct nonlocal electrical measurements to explore the charge-spin interconversion effect. Optical image of a typical device is shown in Supplementary Fig. 3.

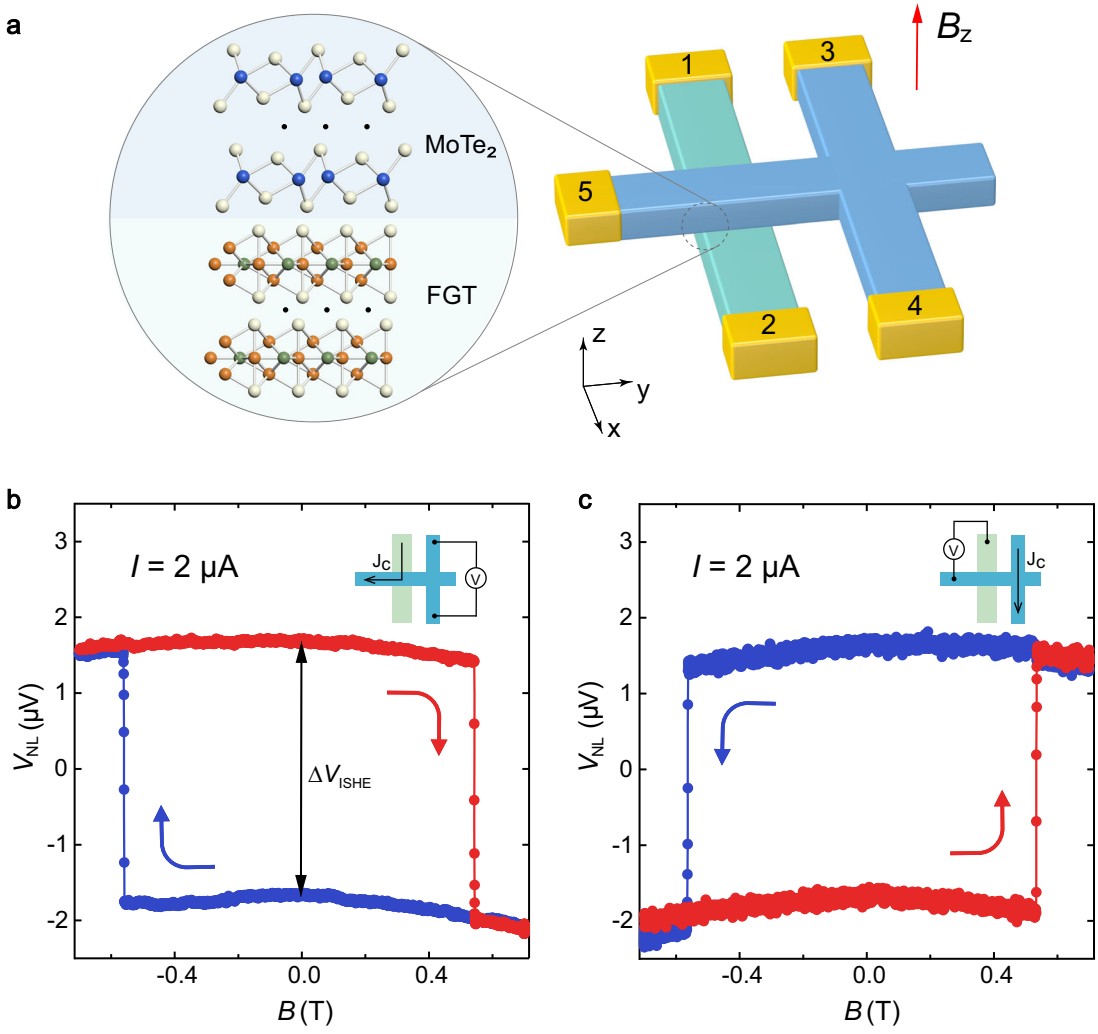

**Fig. 1 | Nonlocal transport measurement configuration and characterization of the SHE. a** Schematic of the FGT/MoTe₂ device for transport measurements. The external magnetic field is swept along z axis. The numbers represent the terminal positions for the measurements. **b** The corresponding inverse spin Hall signal as a function of out-of-plane magnetic field. The voltage jump is denoted as $\Delta V_{ISHE}$. **c** The corresponding spin Hall signal as a function of out-of-plane magnetic field.

To investigate the spin diffusion in $MoTe_2$ used in the vdW heterostructure, we perform an experiment based on non-local measurement scheme[46] where a charge current is applied between terminal 1 on the FGT flake and terminal 5 on $MoTe_2$ flake. In this case, the charge current flows from FGT into $MoTe_2$ and the resulting spin polarization is injected into the $MoTe_2$ and diffuses to the $MoTe_2$ Hall bar (terminals 3 and 4). We monitor the spin diffusion-induced charge imbalance with the voltage signal between terminals 3 and 4, as shown in Fig. 1b. The measurement shows a sharp jump in the voltage signal (denoted as $\triangle V_{ISHE}$) as the out-of-plane magnetization of the FGT is reversed by the applied perpendicular magnetic field. Our results reveal a large non-local inverse spin Hall signal of 1.6 Ω, which is one order of magnitude larger than that measured in local ferromagnetic/heavy metal nanostructures[47]. Such large inverse spin Hall signal indicates that $MoTe_2$ has long spin diffusion length, since the channel length between the heterostructure region and $MoTe_2$ Hall cross (2 $\mu m$) is three orders of magnitude larger than spin diffusion length found in heavy metals[17,48,49]. The long spin diffusion length of $MoTe_2$ can be further verified by our length-dependence transport measure-

ment (Supplementary Note 1). Nevertheless, comprehensive understanding of such a long spin diffusion length observed in $MoTe_2$ with strong spin-orbit coupling requires more theoretical and experimental effort in the future. As shown in Fig. 1c, we applied a charge current between terminals 3 and 4 on $MoTe_2$ to generate a pure spin current along the y direction, resulting in a spin accumulation at the FGT/$MoTe_2$ interface that can be probed by the setup shown in the inset of Fig. 1c. These significant magnetic hysteresis loops of the spin Hall signal and inverse spin Hall signal demonstrate the great potential of using $MoTe_2$ as a highly efficient nonlocal spin detector for exploring the charge-spin conversion at the FGT/$MoTe_2$ heterostructure region.

We then drive a charge current $I_{12}$ between terminals 1 and 2 on the FGT strip and use the nonlocal spin detector to investigate the corresponding charge-spin conversion at the heterostructure region. The nonlocal voltage signal is monitored between Hall terminals 3 and 4 on $MoTe_2$ flake as we vary the magnetic field, with the results shown in Fig. 2a. The measured nonlocal voltage signal displays a rectangular hysteresis loop in response to the applied magnetic field. At a fixed **M** direction (and **B** = 0), the polarity of this hysteresis loop reverses when

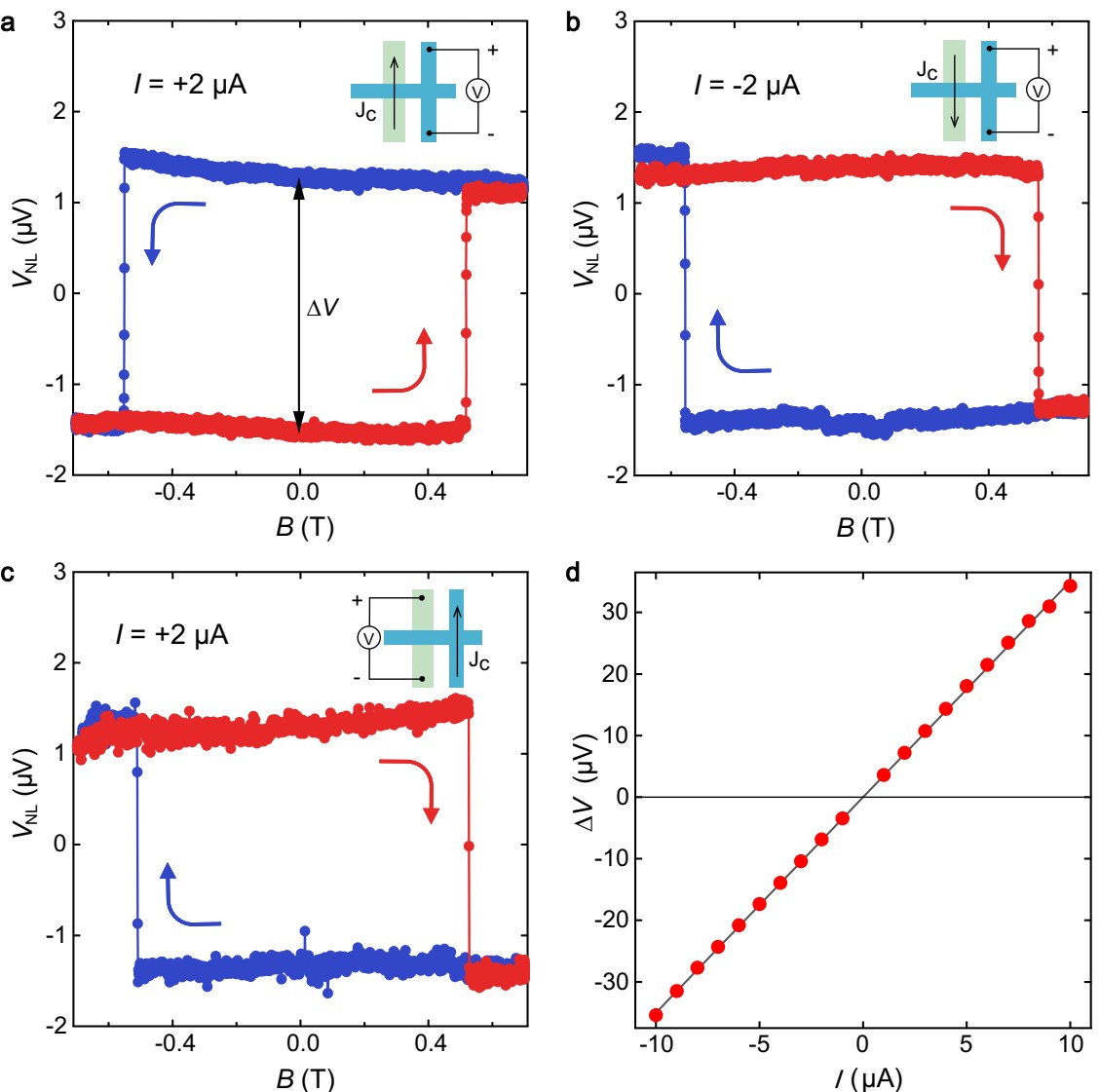

**Fig. 2 | Nonlocal transport measurements of the MSHE. a** The measurement of MSHE with measurement configuration shown in inset. The voltage jump in this $V_{NL}-B$ hysteresis loop is denoted as $\triangle V$. **b** The measurement of MSHE with negative bias current. **c** The measurement of inverse MSHE in the same device. **d** $\triangle V$ as a function of bias current $I$. The line is linear fit to the experiment data.

flipping the direction of the driving current $I_{12}$ (Fig. 2b). It is noteworthy that the sharp jumps in the voltage signal occur at the switching field of FGT, demonstrating the $T$-odd nature of the spin current ($J_y^{S_z}$) induced by the injected charge current along the FGT strip. This implies that the direction of the resulting spin current is flipped when the magnetization of FGT is reversed. This $T$-odd signal is clear evidence for MSHE.

Our device setup is capable of detecting the inverse effect of MSHE. By applying a charge current between terminals 3 and 4 on the MoTe$_2$ flake, depicted in the inset of Fig. 2c, a spin current ($J_y^{S_z}$) is generated through the spin Hall effect in MoTe$_2$. The resulting spin current diffuses from right to left side, and then induces a charge current via the inverse MSHE at the heterostructure region, leading to a $T$-odd voltage signal between terminals 1 and 2 on the FGT flake, as shown in Fig. 2c. The measured voltage signals exhibit a rectangular hysteresis loop with comparable signal magnitude and opposite switching polarity compared to that shown in Fig. 2a, which is consistent with the Onsager reciprocity relation[50].

The observed spin-charge-conversion at the heterostructure region cannot be attributed to the conventional charge-spin conversion mechanisms. The conventional spin Hall effect (whether it is from FGT or from MoTe$_2$ or from the junction) is $T$-even hence cannot lead to this $T$-odd signal. We also make careful analysis to show that Rashba-Edelstein effect has negligible contributions to the observed $T$-odd signal (Supplementary Note 2). In addition, the magnitude $\Delta V$ of this $T$-odd signal exhibits a linear dependence on the driving current $I_{12}$ (Fig. 2d), which eliminates the possibility of thermoelectric contributions in our measurements, as they would result in a quadratic bias current dependence of the resulting voltage signal.

Additionally, we carry out a control experiment by replacing MoTe$_2$ with graphite in the device. As shown in the inset of Supplementary Fig. 4, when a charge current is applied along the FGT strip, no signal jump is observed in the graphite nonlocal detector, even though a background signal is still present. This lack of signal jump can be attributed to the negligible spin-orbit coupling in graphite, which only allows for the detection of charge diffusion rather than spin diffusion in the nonlocal channel. This confirms that the $T$-odd signal observed in our previous measures cannot be explained by the charge diffusion in the MoTe$_2$ channel or anomalous Hall effect and must be a result of a spin-related phenomenon.

**The origin of the interfacial-MSHE**

Our symmetry analysis and calculations reveal that the MSHE in FGT/MoTe$_2$ heterostructure is a result of the interplay between symmetry breaking and band geometric properties at the vdW interface. In our measurement configuration, the ISHE in the MoTe$_2$ is sensitive only to the spin current $J_y^{S_z}$ with spin component out of the transport plane, *i.e.* spin-$z$ polarized[43]. This indicates that $T$-odd spin Hall conductivity (SHC) $\sigma_{yx}^{S_z}$ dominates the MSHE in our device. Based on the rigorous symmetry analysis[51–53], the bulk FGT used in our work is forbidden from having a nonzero $\sigma_{yx}^{S_z}$ due to multiple magnetic symmetries (*e.g.* $C_{2y}T$, $M_xT$, and etc.), as shown in Fig. 3a. However, the relevant symmetry limitations are all broken by stacking the symmetry-mismatched heterointerface formed by FGT and MoTe$_2$, in which the mirror planes of FGT and MoTe$_2$ are misaligned (see more details in Supplementary Note 3). Therefore, this interfacial symmetry breaking allows the nonzero $T$-odd $\sigma_{yx}^{S_z}$, making the observation of the interfacial-MSHE possible in the vdW heterostructure as depicted in Fig. 3b.

Theoretically, the $T$-odd SHC can be described by the equation (the derivation is presented in the Methods section)

$$\sigma_{ab}^{S_c} = \frac{e\tau}{\hbar} D_{ab}^{S_c} = \frac{e\tau}{\hbar} \sum_n \int \frac{d^d k}{(2\pi)^d} f_0 \partial_{k_b} j_{a,n}^{S_c}(\boldsymbol{k}), \tag{1}$$

where $\tau$ is the relaxation time, $j_{a,n}^{S_c}(\boldsymbol{k}) = \langle u_n(\boldsymbol{k})|\frac{1}{2}\{\hat{v}_a, \hat{s}_c\}|u_n(\boldsymbol{k})\rangle$ is the average spin current density for band eigenstate $|u_n(\boldsymbol{k})\rangle$, $\hat{v}_a$ and $\hat{s}_c$ are

the velocity and spin operators, $d$ is the dimension of the system, and $f_0$ is the Fermi distribution function. $D_{ab}^{S_c} = \sum_n \int \frac{d^d k}{(2\pi)^d} f_0 \partial_{k_b} j_{a,n}^{S_c}(\boldsymbol{k})$ can be regarded as a spin current dipole in momentum space, similar to the Berry curvature dipole[54–56]. As $j_{a,n}^{S_c}(\boldsymbol{k})$ is an even function in $k$-space under the time-reversal symmetry, the resulting spin current dipole $D_{ab}^{S_c}$ is an odd function. This behavior is consistent with the ferromagnetic systems and the sign of spin current dipole flips with the magnetic order parameter (see more details in Methods). It is important to note that the spin current dipole is dependent solely on the band structure and only exists in magnetic systems, disappearing in nonmagnetic systems that maintain time-reversal symmetry. This makes the spin current dipole an intrinsic property of each ferromagnetic material.

To evaluate the spin current dipole and the spin Hall conductivity, we use the above equation and conduct the first-principles calculations. Our calculation results, shown in Fig. 3c, reveal a nonzero spin Hall conductivity $\sigma_{yx}^{S_z}$ at the interface of FGT/MoTe$_2$ vdW heterostructure while it vanishes in bulk FGT (see Supplementary Fig. 5). This confirms our early symmetry analysis. The $k$-resolved spin current dipole $D_{yx}^{S_z}$ in the FGT/MoTe$_2$ bilayer system reveals that the spin properties of materials can be manipulated by tailoring the distribution of $J_y^{S_z}$ in momentum space. Additionally, we calculate the temperature dependence of $\sigma_{yx}^{S_z}$ in the FGT/MoTe$_2$ vdW heterostructure by varying the temperature, as shown in Fig. 3d. Our calculations show a monotonic decrease in the spin Hall conductivity with increasing temperature. To validate these results, we also measure the non-local voltage at different temperatures (as shown in Supplementary Fig. 6) by using the configuration shown in the inset of Fig. 2a and extract the temperature dependence of the spin Hall conductivity at the interface of the FGT/MoTe$_2$ heterostructure, by using the following equation

$$\sigma_{SH}^{interface} = \frac{1}{2} B \triangle R_{NL} \frac{1}{\rho_{MoTe_2}^2} \frac{1}{\rho_{FGT}} \frac{1}{\sigma_{SH}}, \tag{2}$$

where $\rho_{MoTe_2}$ and $\rho_{FGT}$ are the resistivity of the MoTe$_2$ and FGT, $\sigma_{SH}$ is the intrinsic spin Hall conductivity of MoTe$_2$, $\triangle R_{NL}$ ($= \Delta V/I$) is the resistance jump of hysteresis loops shown in Supplementary Fig. 6, and $B$ is an interfacial-related constant. Our calculation result (solid line) is in good agreement with the experimental data (symbols), which suggests that the unique interface of vdW heterostructure is responsible for the observed MSHE. In addition, by comparing $\Delta R_{ISHE}$ ($= \Delta V_{ISHE}/I$) and MSHE-related $\Delta R$ measured in the same device, we estimate that the spin Hall angle is 0.67 in our device (see Methods), indicating a highly-efficient charge-spin conversion mechanism. It is noted that our observed magnetic spin Hall signal is highly reproducible and independent of the sample thickness (see Supplementary Note 4 and Supplementary Fig. 7), verifying its intrinsic origin from the spin current dipole of the electronic band structure at the symmetry-mismatched heterointerface.

Based on the above analysis, the interfacial-MSHE observed in the Fig. 2a can be explained as follows. At the interface of the FGT/MoTe$_2$ vdW heterostructure, both time-reversal and spatial-reversal symmetries are broken. This symmetry breaking leads to the tailoring of the band geometry of FGT/MoTe$_2$, causing a redistribution of spin current in $k$-space and resulting in a non-zero $T$-odd transverse spin current. This spin current is generated from the injected charge current in the FGT and flows over the interface. Since the spin current is coherently generated over the interface, it saves the need to cross the interface and suffer from scattering, and generates a large MSHE response. Importantly, this spin current is locked to the magnetization of the FGT, as illustrated in Figs. 3e and 3f. As a result, the direction of generated spin current can be reversed by the applied perpendicular magnetic field, leading to the rectangular hysteresis loop of voltage signal seen in Fig. 2a.

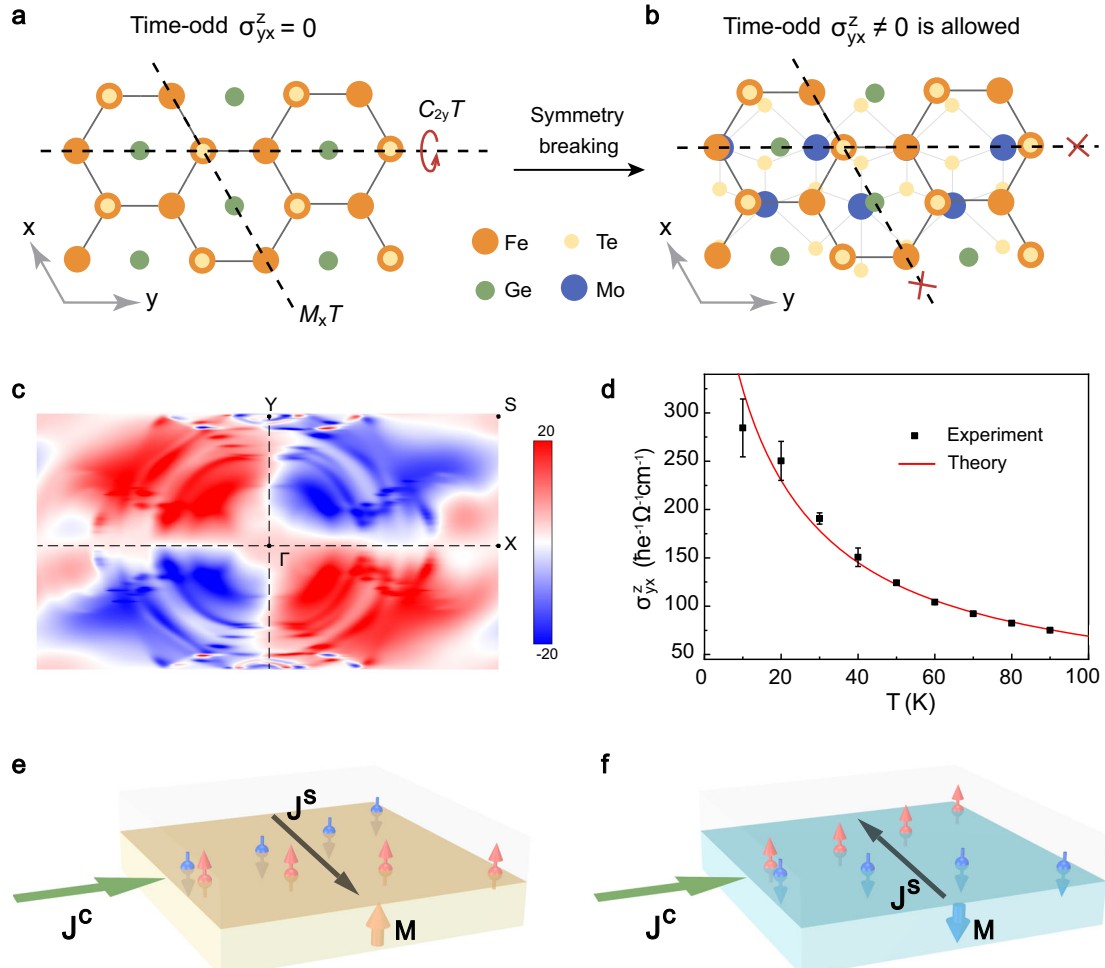

**Fig. 3 | The mechanism for interfacial-MSHE. a** Illustration of crystal structure of bulk FGT with symmetry generator $C_{2y}T$ and $M_xT$, which prohibit $T$-odd SHC $\sigma_{yx}^{S_z}$. **b** Illustration that the $T$-odd SHC $\sigma_{yx}^{S_z}$ is allowed by the symmetry-breaking at the interface of FGT/MoTe$_2$ heterostructure. **c** Calculated $k$-resolved spin current dipole of $\sigma_{yx}^{S_z}$ in FGT/MoTe$_2$ heterostructure. The color code represents the magnitude of the spin current dipole. Different from that in bulk FGT, the distribution of $k$-resolved spin current dipole $D_{yx}^{S_z}$ in FGT/MoTe$_2$ heterostructure results in non-zero integration of spin current dipole and non-zero magnetic spin Hall conductivity. **d** Calculated temperature dependence of $T$-odd SHC $\sigma_{yx}^{S_z}$, which is in good agreement with our experimental results. The calculated τ in our system is

0.04 ps−0.33 ps. The error bars are determined by the noise level of the measured nonlocal voltage signal. **e** Schematics of interfacial-MSHE in a ferromagnetic metal/nonmagnetic metal bilayer. **M** is the magnetization of ferromagnet. $J^S$ is the spin current over the interface. $J^C$ is the applied charge current. The gray box represents the nonmagnetic layer and the orange box represents the magnetic layer with perpendicular magnetization in +z direction. **f** Schematics of the interfacial-MSHE in the same bilayer with the magnetization direction of the magnetic layer reversed. As a time-reversal odd quantity, the direction of th**e** spin current induced by interfacial-MSHE reverses compared to that in **e**.

## Neuromorphic computing based on nonvolatile charge-spin conversion

We propose a proof-of-concept spintronic device, referred to as a "memtransformer", that utilizes the interfacial-MSHE at the junction of the FGT/MoTe$_2$ vdW heterostructure and the inverse spin Hall effect in the MoTe$_2$. The memtransformer, shown in Fig. 4a, operates by using voltage signal as both input ($V^{in}$) and output ($V^{out}$). When the magnetization is pointing upwards, the input voltage signal generates spin current through the MSHE over the vdW interface. The resulting spin current, with its upward polarization, carries the memory information associated with the magnetization and transforms the input voltage into a positive voltage signal, which is detected by the inverse spin Hall effect in the MoTe$_2$. In this way, the spin current acts as a linear, nonvolatile transformer that transforms the input voltage signal into the output voltage signal. This linear, nonvolatile transformation between the input and output electrical signals enables the memtransformer to perform multiplication operation. If the magnetization is switched downwards, the resulting spin current with downward polarization generates a negative voltage output signal, also suitable for multiplication.

Therefore, the memtransformer can be used as a building block for in-memory computing by incorporating it into electrical circuit.

We next conduct the corresponding electrical measurement to validate the operating mechanism of the memtransformer. Figure 4b displays the output voltage ($V^{out}$) as a function of the input voltage ($V^{in}$) in a single memtransformer for both up (square) and down (triangular) magnetization directions. The results show that the output voltage exhibits a strong linear relationship with the input voltage for both magnetization directions, demonstrating that the memtransformer can perform the multiplication, where $V^{out} = w \cdot V^{in}$ and the weight $w$ can be represented by the magnetization (i.e., up magnetization as the $w^+$ and down magnetization as $w^-$). The inset of Fig. 4c shows cascading two memtransformer devices enables to realize multiply accumulate (MAC) operations, which are essential in in-memory computing and artificial intelligence. To demonstrate this application, we used an array of two memtransformer devices to perform vector-vector multiplication, with the corresponding results shown in Fig. 4c. Input voltage $V_i^{in}$ (i = 1,2) is applied onto each device and the accumulated output voltage defined as $V^{out} = w_1 \cdot V_1^{in} + w_2 \cdot V_2^{in}$

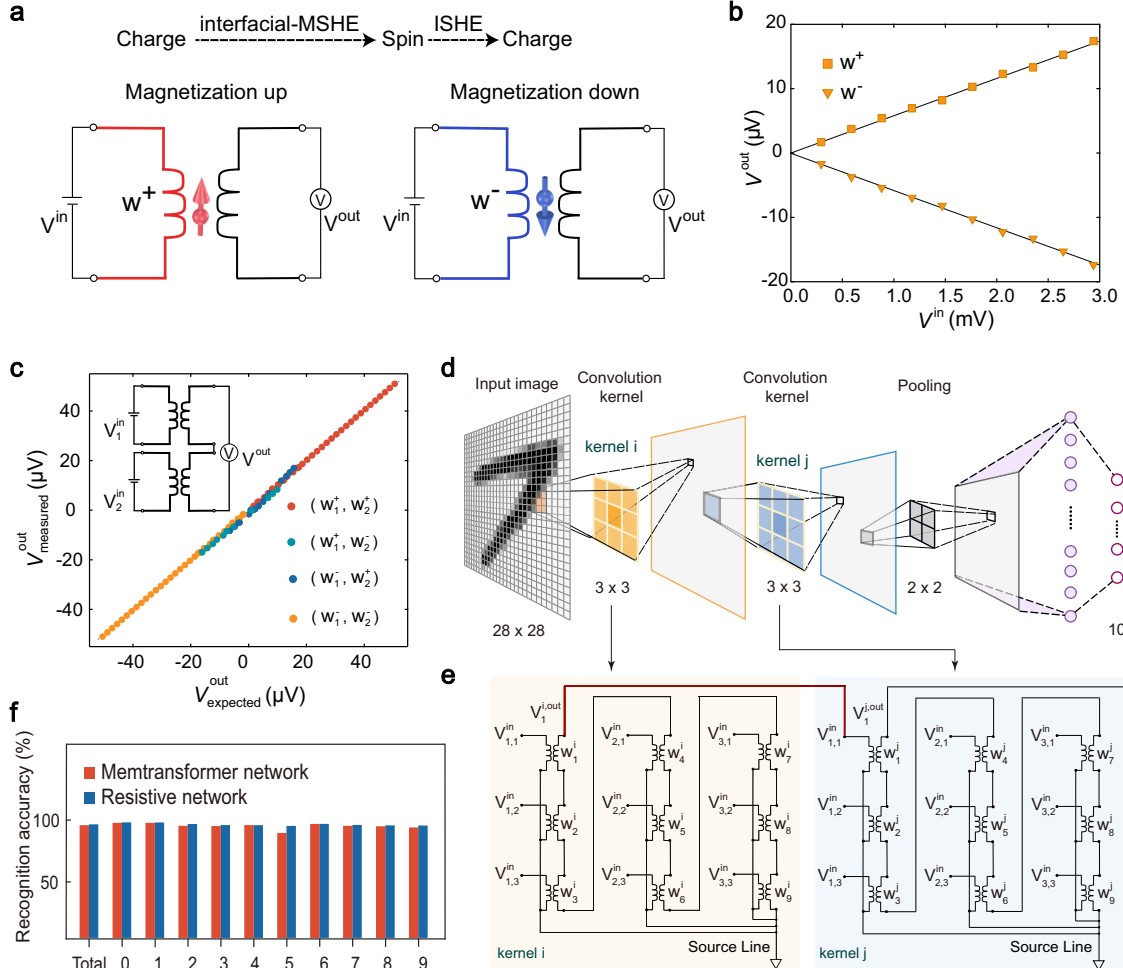

**Fig. 4 | Demonstration of multiply accumulate operation using memtransformer and BCNN for handwritten digit recognition. a** Schematics of memtransformer based on MSHE. **b** $V^{out}$ as a function of $V^{in}$ with positive and negative weights in single memtransformer device. **c** The measured values $V_{measured}^{out}$ as a function of their calculated values $V_{expected}^{out}(= w_1 V_1^{in} + w_2 V_2^{in})$ in array comprised by two memtransformer devices. The weights are denoted as $(w_1^+, w_2^+)$, $(w_1^+, w_2^-)$, $(w_1^-, w_2^+)$, $(w_1^-, w_2^-)$. The inset is schematic of two-memtransformer-based circuits.

**d** Structure of five-layer binary neural network used for MNIST image recognition, with two convolutional layers, one pooling layer and two fully connected layers. **e** The corresponding memtransformer-based arrays of the convolutional kernels in (**d**). **f** Ten thousand MNIST handwritten-digit (0,1,...,9) images are classified with a 96.7% accuracy, which is comparable to recognition accuracy of state-of-the-art binary neural network. Recognition accuracy of the ten different output digits is classified.

is measured, where the weights *i.e.*, $(w_1^+, w_2^+)$, $(w_1^+, w_2^-)$, $(w_1^-, w_2^+)$, $(w_1^-, w_2^-)$ can be changed by switching the magnetization. The experimental result $V_{measured}^{out}$ is in an excellent agreement with the expected result $V_{expected}^{out}$, which is calculated based on the slope shown in the Fig. 4b, indicating that the MAC operations can be successfully implemented with the memtransformer array.

We showcase the potential of the memtransformer in in-memory computing by building a binary convolutional neural network (BCNN), where the binary weights (*i.e.*, −1 and +1) are implemented with the memtransformer. Figure 4d presents a schematic of the BCNN, which consists of two convolutional layers, one pooling layer and two fully connected layers. In the array, the drain terminals of the memtransformer devices are connected to the source lines clamped at 0 V, as depicted in Fig. 4e. We use the BCNN to evaluate the average recognition accuracy of Modified National Institute of Standards and Technology (MNIST) handwritten-digit images and achieve a remarkable accuracy of 96.7% (Fig. 4f), which is comparable to recognition accuracy of state-of-the-art binary neural network. It is worth noting that the two convolutional layers in our proposed BCNN are connected without the use of activation function, indicating that our proposed neural network boasts superior energy

efficiency compared to traditional binary convolutional networks that require activation functions.

## Discussion

In conclusion, we have uncovered an interfacial *T*-odd spin Hall effect in the ferromagnet/metal vdW heterostructure, which has not been identified in the conventional ferromagnet/metal bilayer systems. The *T*-odd nature of this SHE allows for tuning charge-spin conversion by reversing the magnetization, providing a distinct control mechanism for designing spintronic device with new functionalities. As a proof-of-concept, we propose a memtransformer spintronic device and demonstrate its potential for using in binary convolutional neural network. These results not only broaden the field of charge-spin conversion research, but also offer a path towards energy-efficient neuromorphic computing.

## Methods

### Nanofabrication

We cleaved FGT and $MoTe_2$ bulk crystals onto 285 nm $SiO_2$/Si substrates in a glovebox filled with inert atmosphere and selected the exfoliated flakes with the suitable thickness (5−10 nm thick $MoTe_2$ and 25−35 nm thick FGT), with the thickness of these flakes identified with

optical contrast. The exfoliated $MoTe_2$ flakes were then transferred on FGT flakes in inert atmosphere to avoid degradation. E-beam lithography was used to define the electrode pattern, and the electrodes (5 nm Ti/45 nm Au) were deposited by standard electron-beam evaporation. The $MoTe_2$ flakes used in device were then shaped into Hall bar geometry by using standard electron-beam lithography and dry etching in an inductively coupled plasma system. The accurate thicknesses of the $MoTe_2$ and FGT flakes were characterized via a Bruker Multimode atomic force microscope after transport measurements finished. We fabricated $FGT/MoTe_2$ devices at room temperature by using $MoTe_2$ of monoclinic phase (1 T' phase). During the cooling process, $MoTe_2$ of room-temperature monoclinic phase undergoes a structural transition at 250 K, changing into orthorhombic phase ($T_d$ phase) at lower temperature.

### Transport measurements

All the electrical measurements were performed in the Oxford cryostat with magnetic fields of up to 8 T and a base temperature of about 1.6 K. The magnetic field was applied along the out-of-plane direction of the devices. Electrical measurements were performed by lock-in amplifiers (Stanford SR830) using a low-frequency (17.7 Hz) and a Keithley 2636B.

### Calculation details

The electronic structures were carried out in the framework of density functional theory as implemented in the Vienna ab initio simulation package[57,58] with the projector augmented wave method[59] and Perdew, Burke, and Ernzerh of exchange correlation functionals[60]. The result of calculated electronic structures of bulk FGT is shown in Supplementary Fig. 8. For the convergence of the results, the spin–orbit coupling was included self-consistently in the calculations of electronic structures with the kinetic energy cutoff of 700 eV. Since all devices were measured at low temperature, we considered the two-dimensional $FGT/MoTe_2$ bilayer consisted of two layers of FGT and one layer of $T_d$-$MoTe_2$. And a vacuum layer with 15 Å was set to simulate the slab model of $FGT/MoTe_2$ bilayer. Monkhorst-Pack k meshs of $12 \times 12 \times 2$ and $9 \times 5 \times 1$ were used in the FGT bulk and $FGT/MoTe_2$ bilayer, respectively. The s, p orbitals of Te and Ge atoms and s, p, d orbitals of Fe and Mo atoms were used to construct Wannier functions[61]. Based on the real-space Hamiltonian in the maximally-localized Wannier functions basis, we evaluated the spin current dipole and spin conductivity of MSHE within the linear response theory using the Kubo formula, as implemented in the Wannier-Linear-Response code[26]. We also calculated the values of spin conductivity in $FGT/MoTe_2$ heterostructures with varied thicknesses of FGT (Supplementary Fig. 7). The values of T-odd spin Hall conductivity in these systems are independent of the sample thickness, further verifying that the magnetic spin Hall effect observed in $FGT/MoTe_2$ heterostructure is an interfacial effect.

### Calculation of spin Hall angle

The inverse spin Hall signal $\triangle R_{ISHE}$ which is measured with configuration shown in the inset of Fig. 1b can be phenomenologically expressed as[62]:

$$\triangle R_{ISHE} = 2\theta_{SHE} \frac{\rho_{MoTe_2}}{t_{MoTe_2}} P e^{-(L/L_{sf})} \qquad (M1)$$

where $\theta_{SHE}$ is the spin Hall angle of $MoTe_2$, $\rho_{MoTe_2}$ (81.85 $\Omega \cdot nm$) and $t_{MoTe_2}$ (8 nm) are the resistivity and thickness of $MoTe_2$ flake, respectively. $P$ is the effective current spin polarization and $L_{sf}$ is the spin diffusion length of $MoTe_2$. In the calculations of $P$, we use the experimental values of spin Hall angle $\theta_{SHE} = 0.32$ (obtained from measurements of spin Hall effect in $MoTe_2$ in ref. 44) and spin diffusion length $L_{sf} = 1.6$ μm (see Supplementary Note 1), and thus we obtain P ≈ 0.853.

Considering the measurement configuration of MSHE, in which a charge current applied along FGT strip gives rise to a spin current flowing along $MoTe_2$ channel, the non-local voltage jump measured at $MoTe_2$ Hall cross due to the combined MSHE and ISHE can be denoted as[63]:

$$\triangle V_{NL} = 2\rho_{MoTe_2} \theta_{SHE} j_s w \qquad (M2)$$

where $w$ is the width of $MoTe_2$ channel. $j_s$ is the spin current generated by MSHE and can be described as:

$$j_s = j_c \alpha_{MSH} e^{-(L/L_{sf})} \qquad (M3)$$

One can thus obtain the non-local resistance jump measured at $MoTe_2$ Hall cross:

$$\triangle R_{NL} = \frac{\triangle V_{NL}}{I} = 2A\rho_{MoTe_2}\alpha_{MSH}\theta_{SHE}e^{-(L/L_{sf})} \qquad (M4)$$

where $A$ is geometric factor and $\alpha_{MSH}$ is the magnetic spin Hall angle.

By dividing Eq. (M1) and Eq. (M4), we derive the magnetic spin Hall angle:

$$\alpha_{MSH} = \frac{P}{A \cdot t_{MoTe_2}} \cdot \frac{\triangle R_{NL}}{\triangle R_{ISHE}} \qquad (M5)$$

Finally, we obtain the spin Hall angle of about 0.67.

### Analysis of temperature dependence of time-odd spin Hall conductivity originated from the heterostructure region

Measurements of $\triangle R_{NL}$ with configuration shown in the inset of Fig. 2a at different temperatures can be used to derive the temperature dependence of spin Hall conductivity originated from the heterostructure region. From the Eq. (M4), function of the measured non-local resistance jump mentioned above, the spin Hall conductivity of heterostructure region can be expressed as:

$$\sigma_{SH}^{interface} = \frac{1}{2}B\triangle R_{NL} \frac{1}{\rho_{MoTe_2}^2} \frac{1}{\rho_{FGT}} \frac{1}{\sigma_{SHE}} \qquad (M6)$$

where $B$ is an interface-related constant with given geometric parameters and $\sigma_{SHE}$ is the intrinsic spin Hall conductivity of $MoTe_2$ where $\sigma_{SHE} = 176$ $(\hbar/e)\Omega^{-1}cm^{-1}$ (obtained from ref. 43). $\rho_{FGT}$ is the resistivity of FGT. Considering the measured temperature dependence of $\rho_{FGT}$ and $\rho_{MoTe_2}$, the temperature dependence of T-odd spin Hall conductivity can be derived.

### Theory of time-reversal odd spin Hall effect

The spin current generated at the linear order of an applied electric field **E** is given by

$$j_a^c = \sigma_{ab}^c E_b, \qquad (M7)$$

where the Einstein summation convention is adopted for the Cartesian Coordinates, and $j_a^c$ is the current along the $a$ direction of the spin c component.

The spin current density is given by the integral of the spin current carried by each electron $j_n^c(\boldsymbol{k})$ weighted by the distribution function $f_n(\boldsymbol{k})$:

$$j^c = \sum_n \int \frac{d^2k}{(2\pi)^2} f_n(\boldsymbol{k}) j_n^c(\boldsymbol{k}) \qquad (M8)$$

Here n and $\hbar\boldsymbol{k}$ are the band index and crystal momentum, respectively, $j_n^c(k) = \langle u_n(\boldsymbol{k})| \frac{1}{2}\{\hat{v},\hat{s}^c\} |u_n(\boldsymbol{k})\rangle$ and $[d\boldsymbol{k}]$ is shorthand for

$\sum_n d^2\boldsymbol{k}/(2\pi)^2$. In the relaxation time approximation, the deviation from the equilibrium Fermi distribution $f_0(\varepsilon_n)$ is of a dipole structure ($e < 0$):

$$f_n - f_0 = -\frac{e}{\hbar}\tau\boldsymbol{E}\cdot\partial_{\boldsymbol{k}}f_0 \tag{M9}$$

The time reversal odd spin conductivity is at the linear order of the relaxation time τ. The spin conductivity tensor is thus given by

$$\sigma_{ab}^c = \frac{e}{\hbar}\tau\sum_n\int\frac{d^2\boldsymbol{k}}{(2\pi)^2}f_0\partial_{k_b}j_{a,n}^c(\boldsymbol{k}) \tag{M10}$$

where we can introduce

$$D_{ba,n}^c(k) \equiv \partial_{k_b}j_{a,n}^c(k) = \left\langle u_n\left|\frac{1}{2}\left\{\partial_{k_b}\hat{v}_a,\hat{s}^c\right\}\right|u_n\right\rangle$$
$$+ \hbar\sum_{n_1\neq n}\mathrm{Re}\frac{\langle u_n|\{\hat{v}_a,\hat{s}^c\}|u_{n_1}\rangle\langle u_{n_1}|\hat{v}_b|u_n\rangle}{\varepsilon_n - \varepsilon_{n_1}} \tag{M11}$$

We can plot the $k$-space distribution of

$$D_{ba}^c(\boldsymbol{k}) \equiv \sum_n f_0 D_{ba,n}^c(\boldsymbol{k}) = \sum_n f_0\partial_{k_b}j_{a,n}^c(\boldsymbol{k}) \tag{12}$$

### The application of memtransformer array on neuromorphic computing

To show the potential of memtransformer array on achieving large-scale neuromorphic computing, we demonstrate image recognition simulation on a convolution neural network and the MNIST dataset. The MNIST handwritten digit database is composed of 60,000 training images and 10,000 testing images, each with a resolution of $28\times28$ belonging to one of ten digital categories from zero to nine. The neural network has five layers, containing convolution layers, pooling layer and fully connected layers. The network is trained by the stochastic gradient descent for 2000 steps with a batch size of 128, using the Adam optimizer and a cross-entropy loss function. We compare the performance of the neural network with different device models (resistive neural network and memtransformer neural network). The weight of resistive neural network can take any real values without limitation, while that of the memtransformer neural network is bounded to binary values. The recognition accuracy of this memtransformer neural network reached 96.7%, which is comparable to that of resistive neural network, shown in Fig. 4f. It should be noted that in the analog memristor array, its current summation mode would lead to challenges in low-power consumption and large-scale integration due to the finite resistance of crossbar wires. Besides, a neuromorphic computing network generally consists of multiple crossbar arrays, which requires digital-to-analog (analog-to-digital) conversion when large amounts of data go in to (out of) the crossbar arrays, and this also incurs significant costs in area and energy. As such, the cascadable memtransformer array using the voltage sum mode may offer a promising approach to realize energy-efficient and large-scale neuromorphic computing.

### Data availability

The data that support the findings of this study have been presented in the paper and the Supplementary Information. All source data can be acquired from the corresponding authors upon reasonable request. Source data are provided with this paper.

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

## Acknowledgements

This work was supported in part by the National Key R&D Program of China under Grant 2023YFF1203600, the National Natural Science Foundation of China (12322407, 62122036, 62034004, 61921005, 12074176), the Strategic Priority Research Program of the Chinese Academy of Sciences (XDB44000000). F.M. would like to acknowledge support from the AIQ Foundation. The microfabrication center of the National Laboratory of Solid State Microstructures (NLSSM) is acknowledged for their technique support. Crystal growth at Rutgers was supported by the center for Quantum Materials Synthesis (cQMS), funded by the Gordon and Betty Moore Foundation's EPiQS initiative through grant GBMF6402, and by Rutgers University. S.-W.C. acknowledges support by the National Research 389 Foundation of Korea funded by the Ministry of Science and ICT (grant No.2022M3-H4A1A04074153 and 2020M3H4A2084417). C.X. acknowledges support by UM Start-up Grant (SRG2023-00033-IAPME). The work at Hong Kong University was supported by UGC/RGC of Hong Kong SAR (AoE/P-701/20).

## Author contributions

F.M., B.C. and S.-J.L. conceived the idea and supervised the whole project. Y.D. and J.X. fabricated devices and performed transport measurements. L.W., P.W., Z.L. and S.Y. assist the measurements. Y.D., J.X., C.X., B.C., S.A.Y. and S.-J.L. analyzed the data. C.X., Y.G. and S.A.Y did the theoretical analysis and calculations. C.Z. and Y.-G.S. grew $MoTe_2$ bulk crystals. X.X. and S.-W.C. grew $Fe_3GeTe_2$ bulk crystals. Y.D., J.X., B.C., S.-J.L., C.X., S.A.Y. and F.M. wrote the manuscript with input from all authors.

## Competing interests

The authors declare no competing interests.
