## [Peer Review File · Nature Communications]

Reviewers' Comments:

Reviewer #1:

Remarks to the Author:

The submitted paper reports theoretical and experimental study of charge-to-spin conversion (CSC) in Fe₃GeTe₂/MoTe₂ heterostructures which turn out to have an outstanding efficiency and tunability, making it potentially suitable for novel logic-in-memory electronic devices. While I do not deny that such an interface could host intriguing spin-orbit related transport phenomena, the presented research does not seem sufficiently convincing to merit the publication in Nature Communications. The detailed comments are listed below.

1. The state-of-the-art is not properly interpreted. The spin Hall effects in magnetic systems are not new, and they are discussed, for example, in Ref. 66. It is also not true that the 'spintronics devices are limited by symmetry constraints', for example, different and rich configurations of charge-to-spin conversion effects are allowed in low-symmetry crystals, as illustrated in Ref. 38.
2. The authors claim that the effect that they find in FGT/MoTe₂ is novel, but how is it different, in physical sense, from spin Hall effect arising from spin Berry curvature? Is the spin current dipole a new concept proposed by the authors?
3. The symmetry analysis is not properly performed. I would refer here to the earlier papers that provide a symmetry analysis in the group theory language:

<https://journals.aps.org/prb/pdf/10.1103/PhysRevB.92.155138>

<https://journals.aps.org/prb/abstract/10.1103/PhysRevB.92.041101>

<https://journals.aps.org/prmaterials/abstract/10.1103/PhysRevMaterials.6.045004>

What is the space group of the considered heterostructure? What would be the allowed charge-to-spin conversion configurations based on a more rigorous analysis?

4. Can the authors exclude the Rashba-Edelstein effect contributing to the charge-to-spin conversion?

5. What is the physical reason of very long spin diffusion length in MoTe₂?

6. The statement in the conclusions,

'In conclusion, we have uncovered a novel interfacial T-odd spin Hall effect in the ferromagnetic vdW heterostructure. This effect requires only a right symmetry breaking at the interface, making it universally applicable with proper interface engineering.'

is rather vague. Where is the novelty? What is meant by the 'right symmetry'?

8. The DFT calculations are not well connected with the existing literature or the performed experiments. It would be useful to see at least the electronic structure and its agreement with the literature. Also the simulations are performed for FGT/MoTe₂ bilayer while the actual samples have a thickness of a few tens of nanometers.

As stated above, the system is potentially interesting but the research and its analysis have to be performed in a more rigorous way.

Reviewer #2:

Remarks to the Author:

In this manuscript, Yudi Dai et al. report an interfacial magnetic spin Hall effect (interfacial-MSHE) over a Van der Waals (VdW) interface of Fe₃GeTe₂/MoTe₂ heterostructure. Through the non-local electrical transport measurement, the authors have observed a time-reversal-odd charge-spin interconversion effect. Furthermore, the authors perform the detailed theoretical calculations, and

provide a round explanation about interfacial-MSHE, which is attributed to an induced nonzero spin current dipole by breaking both time reversal and spatial inversion symmetry at the vdW interface. Last, the authors propose a model of memtransformer spintronic devices based on the interfacial-MSHE effect. This work is well-organized and informative. It will help readers to expand the research scope of spin-charge interconversion, not only in fundamental research but also in terms of device applications. Therefore, I would like to support this work to be published in Nature Communications subject to satisfactorily addressing the following issues by the authors:

- (1) MoTe₂ could exist in different lattice structures such as 2H, 1T and 1T' phase, it is better to clarify which phase of MoTe₂ is used for this study.
- (2) How many devices have been measured? It seems that the authors have made many devices but they only show data from one device. It will be useful to make a comparison (e.g. the spin Hall angle) with different devices, since the interfacial-MSHE should be related to detailed materials parameters, such as the interface transparency, flake thickness, and so on.
- (3) The spin diffusion length of MoTe₂ is shown to be longer than 2 μm , which is an impressive value as compared to traditional materials. It would be nice to perform a length dependence measurement to get the longest spin diffusion length in this system.

Reviewer #3:

Remarks to the Author:

In this manuscript, the authors reported experimental observations of magnetic spin Hall effect in van der Waals FGT/MoTe₂ bilayers. A neuromorphic computing scheme is also proposed based on the magnetic spin Hall effect. It is an interesting result. However, I have some concerns.

(1) The existence of magnetic spin Hall effect requires breaking all relevant symmetries, including two mirror symmetries normal to the film plane. To my best understanding, FGT alone and MoTe₂ alone do not break those mirror symmetries. So how the interface breaks the symmetry depends on the details of atomic alignment. Therefore, I feel more details about interface structure is needed.

(2) Why would the atoms align the way described in Fig. 3b, and is there an experimental verification? Is the interface structure random or is it the most energy-favorable? Have the authors made more than 1 device? This is important, as for application of computing, many devices have to function the same way.

(3) In addition, I also suggest control experiments for data shown in Fig. 1. For example, in Fig. 1(b), electric current is applied from terminal 1 to terminal 5. Please also apply current from terminal 2 to terminal 5 and see if the voltage signal changes.

(4) It seems that the proposed neuromorphic computing scheme only requires the output magnetization to be depending on input voltage as well as magnetization. I am not an expert in this, but wouldn't the conventional anomalous Hall effect suffice? What is unique about the magnetic spin Hall effect setup, which has lower efficiency in the voltage conversion.

Response to referees' comments

Reviewer #1 (Remarks to the Author):

The submitted paper reports theoretical and experimental study of charge-to-spin conversion (CSC) in Fe₃GeTe₂/MoTe₂ heterostructures which turn out to have an outstanding efficiency and tunability, making it potentially suitable for novel logic-in-memory electronic devices. While I do not deny that such an interface could host intriguing spin-orbit related transport phenomena, the presented research does not seem sufficiently convincing to merit the publication in Nature Communications. The detailed comments are listed below.

Response: We thank the referee for the comments. The emergence of exotic spin-orbit related transport phenomena in the materials is dictated by the symmetry of the material systems. The symmetry of van der Waals (vdW) heterostructure can be readily tailored by choosing different vdW materials and/or stacking orders. In our work, we for the first time observe the interfacial time-reversal-odd (*T*-odd) spin Hall effect (SHE) in the FGT/MoTe₂ vdW heterostructure by engineering the interface symmetry. This is completely different from the well-known SHE observed in traditional ferromagnet/heavy-metal bilayer systems in the following two aspects: 1) The spin current via our interfacial SHE is coherently generated over the interface and saves the need to cross the interface and suffer from scattering, leading to high conversion efficiency. While the generated spin current via conventional SHE, which is *T*-even, must be transmitted through the interface between ferromagnet and heavy-metal, and inevitably suffers from strong interface scattering, resulting in low conversion efficiency. 2) Our interfacial SHE gives rise to a spin current, the direction of which is locked with the magnetization of the ferromagnet. In contrast, the spin current generated through conventional SHE is independent of the magnetization of the ferromagnet. This unique effect allows for designing of a new type of spintronic device, memtransformer, which is promising in neuromorphic computing.

We have confirmed that this novel spin-orbit related interfacial SHE arises from spin current dipole, characterizing the intrinsic geometric band for artificial magnetic heterostructure system. We have ruled out the possibility that this effect is governed by spin Berry curvature-induced SHE or Rashba-Edelstein effect by careful analysis.

Rigorous symmetry analysis and additional thickness-dependence DFT calculations have been performed in the revised manuscript to further verify that our observed SHE is of interfacial effect, instead of bulk effect.

We have fully addressed all the concerns raised by the referee, with the point-to-point responses presented below. We believe that our revised manuscript is sufficiently convincing to merit the publication in Nature Communications.

1. The state-of-the-art is not properly interpreted. The spin Hall effects in magnetic systems are not new, and they are discussed, for example, in Ref. 66. It is also not true that the 'spintronics devices are limited by symmetry constraints', for example, different and rich configurations of charge-to-spin conversion effects are allowed in low-symmetry crystals, as illustrated in Ref. 38.

Response: We thank the referee for the comment. We have added proper interpretation on the state-of-the-art literatures in the revised manuscript. Although the SHE has been discussed in the literatures, theoretically and experimentally, the study of T -odd SHE is rare and only limited in non-collinear bulk antiferromagnet Mn_3Sn and collinear bulk antiferromagnet RuO_2 . Exploring T -odd SHE in ferromagnetic systems, in particular the interfacial effect, would not only scientifically broaden the scope of T -odd SHE, but also technologically provide a distinct route for designing spintronic device with novel functionalities since T -odd SHE in ferromagnet couples the spin Hall current with ferromagnetic order. In our work, by the symmetry engineering at the vdW ferromagnetic interface, we reveal the interfacial T -odd SHE for the first time, and show its promising application in neuromorphic computing.

To avoid any possible confusion, we have added more discussions in the revised manuscript for better interpreting the state-of-the-art research of SHE. For ease of reviewing, the revision has been provided below, which is highlighted in red.

“Coupling the spin-charge conversion with the magnetization in ferromagnet systems with PMA would provide a unique knob for reconfiguring the charge-spin conversion, introducing an additional pathway for developing novel spintronics devices. Recently, it has been proposed that such magnetization-determined charge-spin conversion can be realized through time-reversal-odd SHE^{26,27}, in which the switching of

*magnetization will reverse the spin Hall current. However, magnetic materials hosting such **T**-odd SHE are still rare and only limited in antiferromagnetic systems, i.e., non-collinear bulk antiferromagnet Mn_3Sn ^{18,28} and collinear bulk antiferromagnet RuO_2 ^{20,27,29}. The study of **T**-odd SHE in ferromagnetic systems remains unexplored so far.”* [Line 17, Page 2].

18. Kimata, M. et al. Magnetic and magnetic inverse spin Hall effects in a non-collinear antiferromagnet. *Nature* 565, 627-630 (2019).
20. Bose, A. et al. Tilted spin current generated by the collinear antiferromagnet ruthenium dioxide. *Nat. Electron.* 5, 267-274 (2022).
26. Zelezny, J., Zhang, Y., Felser, C. & Yan, B. Spin-Polarized Current in Noncollinear Antiferromagnets. *Phys. Rev. Lett.* 119, 187204 (2017).
27. Gonzalez-Hernandez, R. et al. Efficient Electrical Spin Splitter Based on Nonrelativistic Collinear Antiferromagnetism. *Phys. Rev. Lett.* 126, 127701 (2021).
28. Kondou, K. et al. Giant field-like torque by the out-of-plane magnetic spin Hall effect in a topological antiferromagnet. *Nat. Commun.* 12, 6491 (2021).
29. Bai, H. et al. Observation of Spin Splitting Torque in a Collinear Antiferromagnet RuO_2 . *Phys. Rev. Lett.* 128 (2022).

The sentence “**spintronic devices are limited by symmetry constraints**” in our original manuscript may mislead the referee, it refers to the fact that the configuration of charge-to-spin conversion is dictated by the symmetry of the materials. This is because the spintronic devices are based on charge-to-spin conversion, while the specific configuration of charge-to-spin conversion is restricted by the symmetry of materials. In this sense, the functionalities of spintronic devices are limited by the materials with specific lattice symmetry. To avoid misleading the authorship of interest, we have revised this sentence in the revised manuscript. For ease of reviewing, the revision has been provided below, which is highlighted in red.

“Notably, such symmetry engineering at vdW heterointerface plays a vital role in generating unconventional charge-spin conversion mechanisms, such as non-orthogonal SHE for realizing field-free current-induced perpendicular magnetization reversal³⁴.” [Line 33, Page 2].

34. MacNeill, D. et al. Control of spin-orbit torques through crystal symmetry in WTe_2 /ferromagnet bilayers. *Nat. Phys.* 13, 300-305 (2016).

2. The authors claim that the effect that they find in FGT/MoTe₂ is novel, but how is it different, in physical sense, from spin Hall effect arising from spin Berry curvature? Is the spin current dipole a new concept proposed by the authors?

Response: We thank the referee for this comment. The SHE observed in FGT/MoTe₂ heterostructure in our work is different from the conventional spin Hall effect arising from spin Berry curvature in the two aspects: 1) Our interfacial SHE is time-reversal-odd and arises from the spin current dipole, while the conventional SHE is time-reversal-even and can be attributed to spin Berry curvature; 2) In our work, the spin current via the T -odd SHE is locked to the magnetization direction of ferromagnet. While the spin current generated via T -even SHE is independent of the magnetization direction of ferromagnet.

We introduce the spin current dipole (which is a band geometric quantity) in our work, to well describe the physics of our interfacial T -odd SHE. This terminology is introduced in the same spirit as the Berry curvature dipole [Phys. Rev. Lett. 115, 216806 (2015), Nature 565, 337 (2019)]. Such band geometric quantity can be conveniently used as key index for labelling the physical content of the T -odd effect in more materials.

3. The symmetry analysis is not properly performed. I would refer here to the earlier papers that provide a symmetry analysis in the group theory language:

<https://journals.aps.org/prb/pdf/10.1103/PhysRevB.92.155138>

<https://journals.aps.org/prb/abstract/10.1103/PhysRevB.92.041101>

<https://journals.aps.org/prmaterials/abstract/10.1103/PhysRevMaterials.6.045004>

What is the space group of the considered heterostructure? What would be the allowed charge-to-spin conversion configurations based on a more rigorous analysis?

Response: We thank the referee for the helpful suggestion and for bringing these valuable references to our attention. By following the suggestion, we have carried out the rigorous symmetry analysis based on the Neumann's principle, and confirmed that

the FGT/MoTe₂ heterointerface belongs to the Laue group of -1. Because we did not align the mirrors of these two materials in the stacking process, no mirror plane is preserved in the heterointerface. In this case, nine elements in T -odd part of the spin Hall conductivity tensor σ^z , including σ_{yx}^z , σ_{xz}^z , σ_{yz}^z , are all allowed to have non-zero values. To reflect the reviewer's suggestions, we have added these literatures mentioned by the referee into our revised manuscript (see Ref. 50-52), and have added more rigorous symmetry analysis in the main text. For ease of reviewing, the revision has been provided below, which is highlighted in red.

“Based on the rigorous symmetry analysis⁵⁰⁻⁵², the bulk FGT used in our work is forbidden from having a nonzero σ_{yx}^{Sz} due to multiple magnetic symmetries (e.g. $C_{2y}T$, M_xT , and etc.), as shown in Fig. 3a. However, the relevant symmetry limitations are all broken by stacking the symmetry-mismatched heterointerface formed by FGT and MoTe₂, in which the mirror planes of FGT and MoTe₂ are misaligned (see more details in Supplementary Note 3). Therefore, this interfacial symmetry breaking allows the nonzero T -odd σ_{yx}^{Sz} , making the observation of the interfacial-MSHE possible in the vdW heterostructure as depicted in Fig.3b.” [Line 5, Page 6].

50. Seemann, M., Ködderitzsch, D., Wimmer, S. & Ebert, H. Symmetry-imposed shape of linear response tensors. *Phys. Rev. B* 92, 155138 (2015).

51. Wimmer, S., Seemann, M., Chadova, K., Ködderitzsch, D. & Ebert, H. Spin-orbit-induced longitudinal spin-polarized currents in nonmagnetic solids. *Phys. Rev. B* 92, 041101 (2015).

52. Roy, A., Guimarães, M. H. D. & Sławińska, J. Unconventional spin Hall effects in nonmagnetic solids. *Physical Review Materials* 6, 045004 (2022).

4. Can the authors exclude the Rashba-Edelstein effect contributing to the charge-to-spin conversion?

Response: We thank the referee for this valuable question. We have performed a careful analysis to exclude the contribution of Rashba-Edelstein effect to the charge-to-spin conversion observed in our work. The specific analysis is summarized below:

- (1) Rashba-Edelstein effect can only induce in-plane nonequilibrium spin density and the resulting spin current cannot be detected by the MoTe₂ used in our work,

which is consistent with our experimental results, as shown in Fig. 2a of our revised manuscript. According to Rashba-Edelstein effect, when we apply a charge current along x-axis in heterostructure, the spins generated from Rashba effect is polarized along y-axis (i.e., S_y) and can diffuse along y-axis, denoted as $j_y^{S_y}$. MoTe₂-based non-local spin detector (with the detection Hall bar along x-axis) used in our FGT/MoTe₂ device cannot detect the resulting spin current, instead can only detect the spin current along y-axis with spin polarization along z-axis (corresponding to σ_{yx}^z) via inverse spin Hall effect. This is because T_d-MoTe₂ only allows six nonzero spin Hall conductivity tensor elements with orthogonal charge current, spin current and spin polarization [see previous theoretical work in Phys. Rev. B 99, 060408 (2019)], as shown in Table R1. This result indicates that the Rashba-Edelstein effect cannot contribute to the charge-to-spin conversion observed in our work.

Table R1 Symmetry-allowed spin Hall conductivity tensors of T_d-MoTe₂

	σ^x	σ^y	σ^z
Space Group Pmn2 ₁	$\begin{pmatrix} 0 & 0 & 0 \\ 0 & 0 & \sigma_{yz}^x \\ 0 & \sigma_{zy}^x & 0 \end{pmatrix}$	$\begin{pmatrix} 0 & 0 & \sigma_{xz}^y \\ 0 & 0 & 0 \\ \sigma_{zx}^y & 0 & 0 \end{pmatrix}$	$\begin{pmatrix} 0 & \sigma_{xy}^z & 0 \\ \sigma_{yx}^z & 0 & 0 \\ 0 & 0 & 0 \end{pmatrix}$

The spin Hall conductivity tensor $\sigma_{\alpha\beta}^\gamma$ is a third-order tensor, where β and α are the direction of charge current and spin current, respectively. γ is spin polarization.

(2) The spin current generated over FGT/MoTe₂ heterostructure region in our device is T -odd, and thus the direction of generated spin current is locked to the magnetization of FGT. However, the Rashba effect is only determined by the SOC and inversion symmetry breaking at the interface, and is irrelevant to the direction of magnetization. Thus, the T -odd spin current in our work cannot be induced by Rashba effect.

To remove any possible confusion, we have revised the manuscript and added the above discussion into Supplemental Information (see Supplementary Note 2). For ease of reviewing, the revision has been provided below, which is highlighted in red.

“We also make careful analysis to show that Rashba-Edelstein effect has negligible contributions to the observed T-odd signal (Supplementary Note 2).” [Line 16, Page 5].

5. What is the physical reason of very long spin diffusion length in MoTe₂?

Response: We thank the referee for this question. The long spin diffusion length observed in our MoTe₂ device is consistent with the results reported in the previous work [Nature Materials 19, 292 (2020)]. The specific physical reason remains an open question and calls for more theoretic efforts. However, several possible reasons, including Dyakonov-Perel-type spin relaxation mechanism, complex band structure (including intricate spin texture and rich orbital texture), have been proposed in previous work to have an understanding of such long spin diffusion length. Since our work mainly focuses on the experiment, understanding of this long spin diffusion length goes beyond the scope of this work and will be studied in the next work.

6. The statement in the conclusions, 'In conclusion, we have uncovered a novel interfacial T-odd spin Hall effect in the ferromagnetic vdW heterostructure. This effect requires only a right symmetry breaking at the interface, making it universally applicable with proper interface engineering.' is rather vague. Where is the novelty? What is meant by the 'right symmetry'?

Response: We thank the referee for the suggestion. By following the useful suggestion, we have modified the statement in the conclusions in the revised manuscript, so that the novelty of our work can be presented in a clear manner. For ease of reviewing, we have provided the revision in the following, which has been highlighted in red.

“In conclusion, we have uncovered a new interfacial T-odd spin Hall effect in the ferromagnet/metal vdW heterostructure, which has not been identified in the conventional ferromagnet/metal bilayer systems. The T-odd nature of this SHE allows for tuning charge-spin conversion by reversing the magnetization, providing a distinct control mechanism for designing spintronic device with new functionalities. As a proof-of-concept, we propose a memtransformer spintronic device and demonstrate its potential for using in binary convolutional neural network. These results not only

broaden the field of charge-spin conversion research, but also offer a path towards energy-efficient neuromorphic computing.” [Line 24, Page 9].

The phrase ‘right symmetry’ refers to the intentional symmetry engineering in vdW heterointerface. Without the constraint of lattice mismatch, the interface symmetry of the vdW heterostructure can be engineered by intentionally choosing distinct materials with distinct symmetry. In this sense, such interfacial symmetry engineering allows for achieving specific symmetry at the heterointerface and revealing previously unidentified exotic spin-orbit related phenomena, thus promising as a universal strategy. To exclude any possible confusion to the readership of interest, we have deleted this word “right symmetry” in the revised manuscript. For ease of reviewing, the revision has been provided below, which is highlighted in red.

“In conclusion, we have uncovered a new interfacial T-odd spin Hall effect in the ferromagnet/metal vdW heterostructure, which has not been identified in the conventional ferromagnet/metal bilayer systems. The T-odd nature of this SHE allows for tuning charge-spin conversion by reversing the magnetization, providing a distinct control mechanism for designing spintronic device with new functionalities. As a proof-of-concept, we propose a memtransformer spintronic device and demonstrate its potential for using in binary convolutional neural network. These results not only broaden the field of charge-spin conversion research, but also offer a path towards energy-efficient neuromorphic computing.” [Line 24, Page 9].

7. The DFT calculations are not well connected with the existing literature or the performed experiments. It would be useful to see at least the electronic structure and its agreement with the literature. Also the simulations are performed for FGT/MoTe₂ bilayer while the actual samples have a thickness of a few tens of nanometers.

Response: We thank the referee for the valuable suggestions, which have helped us improve our manuscript. Following the referee’s suggestion, we have added the electronic band structure calculations of bulk FGT (see Fig. R1) and the calculations of interfacial-MSHE in FGT/MoTe₂ heterostructures with different sample thicknesses (see Fig. R2) in the revised manuscript (Supplementary Figure 7, 8). Fig. R1 demonstrates our calculated band structure of bulk FGT, and it is consistent with the

previous works [Nature Materials. 17, 794 (2018); Phys. Rev. B 105, 014437 (2022)]. Figure R2 shows the calculated Γ (defined as $\hbar/2\tau$, with τ the relaxation time) dependence of the T -odd spin Hall conductivity in heterostructures stacked by single layer MoTe₂ and two/three/four/five layers FGT. We found that the values of spin Hall conductivity in heterostructures with varied thicknesses of FGT are comparable. Moreover, the calculated Γ -dependence of the T -odd spin Hall conductivity in these four systems are independent of the thickness, and consistent with our temperature-dependent experimental results, indicating that the magnetic spin Hall effect observed in our work is an interfacial effect and well connected with our interfacial symmetry analysis.

Fig. R1. Calculated electronic structures with spin-orbit coupling (SOC) of bulk FGT in the DFT calculation (black lines) and the interpolation with maximally localized Wannier functions (MLWFs) (red dashed lines).

Fig. R2. Calculated Γ dependence of T -odd spin Hall conductivity σ_{yx}^z in heterostructures with different thicknesses of FGT.

As stated above, the system is potentially interesting but the research and its analysis have to be performed in a more rigorous way.

Response: By following the reviewer's helpful suggestions, we have added detailed discussions on the differences between interfacial T -odd effect we observed and other effects (e.g., T -even SHE arising from spin Berry curvature and Rashba-Edelstein effect), rigorous symmetry analysis of heterostructure and additional thickness-dependence DFT calculations into our revised manuscript. We hope that the reviewer will find our revised version suitable for being published in Nature Communications.

Reviewer #2 (Remarks to the Author)

In this manuscript, Yudi Dai et al. report an interfacial magnetic spin Hall effect (interfacial-MSHE) over a Van der Waals (VdW) interface of Fe₃GeTe₂/MoTe₂ heterostructure. Through the non-local electrical transport measurement, the authors have observed a time-reversal-odd charge-spin interconversion effect. Furthermore, the authors perform the detailed theoretical calculations, and provide a round explanation about interfacial-MSHE, which is attributed to an induced nonzero spin current dipole by breaking both time reversal and spatial inversion symmetry at the vdW interface. Last, the authors propose a model of memtransformer spintronic devices based on the interfacial-MSHE effect. This work is well-organized and informative. It will help readers to expand the research scope of spin-charge interconversion, not only in fundamental research but also in terms of device applications. Therefore, I would like to support this work to be published in Nature Communications subject to satisfactorily addressing the following issues by the authors:

Response: We thank the referee for highly appreciating the novelty of our work, and recommending it for publication. By carrying out new experiments and adding new discussions in the revised manuscript, we have fully addressed all the questions raised by the referee. The point-by-point responses are listed below.

(1) MoTe₂ could exist in different lattice structures such as 2H, 1T and 1T' phase, it is better to clarify which phase of MoTe₂ is used for this study.

Response: We thank the referee for the valuable suggestion. T_d-phase MoTe₂ was used in our study. According to the literature [Nature Communications 7, 13552 (2016)], MoTe₂ exhibits monoclinic phase (1T' phase) at room-temperature, and undergoes a structural phase transition changing to orthorhombic phase (T_d phase) below 250 K. Note that all the experiments in our study were performed at temperature below 250 K. Therefore, MoTe₂ used in our work exhibits T_d-phase. We have followed the reviewer's suggestion to add the information of the phase of MoTe₂ in the revised manuscript (see Methods).

(2) How many devices have been measured? It seems that the authors have made many devices but they only show data from one device. It will be useful to make a comparison

(e.g., the spin Hall angle) with different devices, since the interfacial-MSHE should be related to detailed material parameters, such as the interface transparency, flake thickness, and so on.

Response: We thank the referee for the constructive suggestion. We have carried out measurements in 5 distinct devices. In Fig. R3, we have provided the experimental magnetic spin Hall signals measured in these five different devices (Device #1-Device #5). Obviously, we found that all the devices exhibit similar behaviors of magnetic SHE.

Fig. R3. Nonlocal transport measurements of the interfacial-MSHE in five different devices (Device #1-Device #5). The resistance jumps in the R_{NL} - B hysteresis loop are denoted as ΔR_{NL} .

To make a direct comparison, we list the measured magnetic spin Hall signals and corresponding geometrical parameters in these devices, as shown in Table R2, where t_{FGT} , t_{MoTe_2} and L are the thickness of FGT, the thickness of MoTe₂ and the channel length between the FGT/MoTe₂ heterostructure region and MoTe₂ Hall cross. ΔR_{NL} and α are the non-local signal and the calculated magnetic spin Hall angle, respectively. It is noted that the significant magnetic spin Hall signals can be obtained in different

devices, all of which are larger than spin Hall signal (0.3Ω) measured in local ferromagnetic/heavy metal nanostructures [Nature Electron 3, 309 (2020)].

Table R2 Summary of devices and devices parameters

Sample	t_{FGT} (nm)	t_{MoTe_2} (nm)	L (μm)	ΔR_{NL} (Ω)	α
Device 1	25.6	5.4	1.67	1.9	0.74
Device 2	30.8	9.4	2	1.4	0.67
Device 3	25	9.2	2.2	0.85	0.45
Device 4	29.1	8	2.98	0.54	0.48
Device 5	30	8.5	3.08	0.48	0.45

To reflect the reviewer's suggestions, we have added the new experimental results and detailed discussion in the revised manuscript and Supplementary Information [see Supplementary Note 4]. For ease of reviewing, we have provided the revision below, which is highlighted in red.

“It is noted that our observed magnetic spin Hall signal is highly reproducible and independent of the sample thickness (see Supplementary Note 4 and Supplementary Figure 7), verifying its intrinsic origin from the spin current dipole of the electronic band structure at the symmetry-mismatched heterointerface.” [Line 21, Page 7]

(3) The spin diffusion length of MoTe₂ is shown to be longer than 2 μm , which is an impressive value as compared to traditional materials. It would be nice to perform a length dependence measurement to get the longest spin diffusion length in this system.

Response: We thank the referee for this helpful suggestion. Following this suggestion, we have carried out length-dependence transport experiments and obtained a spin diffusion length of 1.6 μm in MoTe₂. Figure R4 shows the length-dependent R_{SH} and a fit to $R_{\text{SH}} \propto e^{-L/L_{\text{sf}}}$ [by using the model in Phys. Rev. B. 79, 035304 (2009)]. The value we obtained is comparable with that (2.2 μm) reported in few-layer MoTe₂ [Nature Materials 19, 292–298 (2020)]. To remove any confusion, we have revised the manuscript and added the experimental results in Supplementary Information [see

Supplementary Note 1]. For ease of reviewing, the revision has been provided below, which is highlighted in red.

“Our results reveal a large non-local inverse spin Hall signal of 1.6Ω , which is one order of magnitude larger than that measured in local ferromagnetic/heavy metal nanostructures⁴⁷. Such large inverse spin Hall signal indicates that MoTe₂ has long spin diffusion length, since the channel length between the heterostructure region and MoTe₂ Hall cross ($2 \mu\text{m}$) is three orders of magnitude larger than spin diffusion length found in heavy metals^{17,48,49}. The long spin diffusion length of MoTe₂ can be further verified by our length-dependence transport measurement (Supplementary Note 1)” [Line 10, Page 4].

17. Liu, L. et al. Spin-Torque Switching with the Giant Spin Hall Effect of Tantalum. *Science* 336, 555-558 (2012).

47. Pham, V. T. et al. Spin-orbit magnetic state readout in scaled ferromagnetic/heavy metal nanostructures. *Nat. Electron.* 3, 309-315 (2020).

48. Pai, C.-F. et al. Spin transfer torque devices utilizing the giant spin Hall effect of tungsten. *Appl. Phys. Lett.* 101, 122404 (2012).

49. Kimura, T., Otani, Y., Sato, T., Takahashi, S. & Maekawa, S. Room-temperature reversible spin Hall effect. *Phys. Rev. Lett.* 98, 156601 (2007).

Fig. R4. Length-dependence R_{SH} at 1.5 K measured by configuration shown in the inset. L is the center-to-center distance between current injectors and voltage probes. The polarization direction of spin illustrated in inset is perpendicular to the transport plane.

Reviewer #3 (Remarks to the Author)

In this manuscript, the authors reported experimental observations of magnetic spin Hall effect in van der Waals FGT/MoTe₂ bilayers. A neuromorphic computing scheme is also proposed based on the magnetic spin Hall effect. It is an interesting result. However, I have some concerns.

Response: We thank the reviewer for the positive evaluation that our result is interesting. We have fully addressed all the concerns. The detailed point-by-point replies are listed below.

(1) The existence of magnetic spin Hall effect requires breaking all relevant symmetries, including two mirror symmetries normal to the film plane. To my best understanding, FGT alone and MoTe₂ alone do not break those mirror symmetries. So how the interface breaks the symmetry depends on the details of atomic alignment. Therefore, I feel more details about interface structure is needed.

Response: We thank the referee for the constructive suggestion. We have added more details about the interface structure in the revised manuscript. The bulk FGT belongs to the magnetic point group $6/m\bar{m}'m'$, while bulk T_d-MoTe₂ belongs to the space group Pmn2₁ (no. 31) with two mirror symmetries, *i.e.*, a pure mirror M_a and a glide mirror \bar{M}_b (Fig. R5). When those two materials are stacked together, the glide mirror is not preserved at the interface of FGT/MoTe₂ heterostructure. The intentional or non-intentional alignment between the mirror planes of both FGT and MoTe₂ will determine the existence of T -odd spin Hall conductivity σ_{yx}^z . When the mirror planes of both FGT and MoTe₂ are not intentionally aligned, the heterointerface has no mirror symmetry and the resulting symmetry belongs to the Laue group of -1. In this way, all relevant symmetries are broken and nonzero T -odd spin Hall conductivity σ_{yx}^z is allowed, as shown in our work. While the mirror planes of both FGT and MoTe₂ are intentionally aligned, the mirror plane perpendicular to the film plane is preserved and the heterostructure symmetry belongs to $2'/m'$, which forbids the T -odd spin Hall conductivity σ_{yx}^z .

Fig. R5. Crystal symmetry of bulk FGT and MoTe₂. **a**, Crystal structure of bulk FGT with symmetry generators, *e. g.*, $C_{2y}T$ and M_xT . **b**, Crystal structure of bulk MoTe₂ with mirror symmetry M_a .

To reflect the reviewer’s suggestion, we have revised the manuscript and added detailed discussions on the interface structure in the main text and Supplementary Information. For ease of reviewing, the revision has been provided below, which is highlighted in red.

*“Based on the rigorous symmetry analysis⁵⁰⁻⁵², the bulk FGT used in our work is forbidden from having a nonzero $\sigma_{yx}^{S_z}$ due to multiple magnetic symmetries (*e.g.* $C_{2y}T$, M_xT , and *etc.*), as shown in Fig. 3a. However, the relevant symmetry limitations are all broken by stacking the symmetry-mismatched heterointerface formed by FGT and MoTe₂, in which the mirror planes of FGT and MoTe₂ are misaligned (see more details in Supplementary Note 3). Therefore, this interfacial symmetry breaking allows the nonzero **T-odd** $\sigma_{yx}^{S_z}$, making the observation of the interfacial-MSHE possible in the vdW heterostructure as depicted in Fig. 3b.” [Line 5, Page 6].*

50. Seemann, M., Ködderitzsch, D., Wimmer, S. & Ebert, H. Symmetry-imposed shape of linear response tensors. *Phys. Rev. B* 92, 155138 (2015).

51. Wimmer, S., Seemann, M., Chadova, K., Ködderitzsch, D. & Ebert, H. Spin-orbit-induced longitudinal spin-polarized currents in nonmagnetic solids. *Phys. Rev. B* 92, 041101 (2015).

52. Roy, A., Guimarães, M. H. D. & Sławińska, J. Unconventional spin Hall effects in nonmagnetic solids. *Physical Review Materials* 6, 045004 (2022).

(2) Why would the atoms aligned the way described in Fig. 3b, and is there an experimental verification? Is the interface structure random or is it the most energy-favorable? Have the authors made more than 1 device? This is important, as for application of computing, many devices have to function the same way.

Response: We thank the referee for the comments. In our experiment, we did not intentionally align the FGT and MoTe₂. The atomic structure shown in Fig. 3b is just used as a schematic for the interfacial symmetry breaking. According to rigorous symmetry analysis made in the **Response #1**, the FGT/MoTe₂ heterointerface belongs to the Laue group of -1 as long as the mirror planes of FGT and MoTe₂ are not aligned. The stacked heterostructure is stable once fabricated and won't change or relax. In our experiments, we have fabricated 5 devices, and carried out the corresponding electrical transport on these 5 devices, with the results shown in Fig. R6. The results demonstrate that these devices exhibit similar behaviors of magnetic spin Hall effect, showing promising applications in neuromorphic computing.

Fig. R6. Nonlocal transport measurements of the interfacial-MSHE in five different devices (Device #1-Device #5). The resistance jumps in the $R_{NL}-B$ hysteresis loop are denoted as ΔR_{NL} .

(3) In addition, I also suggest control experiments for data shown in Fig. 1. For example, in Fig. 1(b), electric current is applied from terminal 1 to terminal 5. Please also apply current from terminal 2 to terminal 5 and see if the voltage signal changes.

Response: We thank the referee for the valuable suggestion. Following this suggestion, we have carried out control experiment by injecting electric current from terminal 2 to terminal 5 (as shown in left panel of Fig. R7). We observed that the polarity of the measured R_{NL} - B hysteresis loops is same as that case of applying current from terminal 1 to terminal 5 (as shown in right panel of Fig. R7). The reason can be explained as follows: According to the measurement configuration, spin polarized current is injected perpendicularly from FGT into MoTe₂, resulting in pure spin current diffusing from heterostructure region to MoTe₂ Hall cross. An induced charge imbalance is measured between Hall cross via inverse SHE. Since the spin polarization of the injected spin current only depends on the magnetization of FGT, the directions of sharp jump in non-local signal are unchanged under these two different configurations of current injection. We also note that there is a slightly deviation of the value of non-local signals in the control experiments, which is due to a nonvanishing in-plane charge current distribution at the FGT/MoTe₂ interfacial region.

Fig. R7. Nonlocal transport measurements of inverse SHE under different measurement configurations. **a**, The inverse spin Hall signal as a function of out-of-plane magnetic field, in which charge current is applied from terminal 2 to terminal 5. The measurement configuration is shown as the inset, where the numbers mark the terminals. The green

strip represents FGT and the blue Hall cross represents MoTe₂. **b**, The inverse spin Hall signal measured by applying charge current from terminal 1 to terminal 5.

(4) It seems that the proposed neuromorphic computing scheme only requires the output magnetization to be depending on input voltage as well as magnetization. I am not an expert in this, but wouldn't the conventional anomalous Hall effect suffice? What is unique about the magnetic spin Hall effect setup, which has lower efficiency in the voltage conversion.

Response: We thank the referee for this question. The proposed neuromorphic computing scheme strongly depends on the use of two integrated interfacial T-odd SHE devices based on the heterostructure. For the AHE-based integrated array, charge current is applied along the longitudinal direction, and the resulting transverse charge current induced via AHE is sensed to probe the magnetic states. Since the Hall bar device is a four-terminal device in which every terminal is conductive, there are additional paths where the leakage charge current can flow in cascaded AHE-based devices, as shown schematically in Fig. R8a. The leakage charge current causes interference between adjacent devices and thus gives rise to an error when reading the state of the device. In contrast to the AHE-based device, the memtransformer proposed in our work uses spin current generated from interfacial-MSHE to carry the memory information. Such spin current diffuses and then is converted into transverse output charge signal via non-local approach, which solves the leakage charge current issue (Fig. R8b).

As for conversion efficiency, the voltage conversion efficiency (0.21) of memtransformer is comparable with the largest anomalous Hall angle (0.2) measured in ferromagnetic Weyl semimetals Co₃Sn₂S₂ [Nature Physics 14, 1125 (2018)]. Note that the memtransformer is a proof-of-concept in our work, its voltage conversion efficiency can be further improved by optimizing the material selection and device fabrication process, which will be done in our future work.

Fig. R8. Illustration of AHE- and memtransformer-based array. **a,** In AHE-based array, leakage charge current can flow through adjacent device, which leads to an error in the readout of magnetic state. **b,** In memtransformer-based array, spin current generated via interfacial-MSHE is used to carry the information of magnetization, which avoids the influence of leakage charge current.

Reviewers' Comments:

Reviewer #1:

Remarks to the Author:

The authors have diligently addressed the comments provided by the referees and have made significant revisions to the paper, accompanied by a comprehensive response letter. As highlighted in the previous review, the findings presented in this work are both novel and interesting, making them potentially suitable for publication in Nature Communications. However, I remain concerned about the introduction of the theory presented in Eq. 1 and the subsequent discussion.

While the authors have correctly evaluated Eq. 1 based on Wannier parametrization, my primary concern lies with Eq. 1 itself. The authors claim to define a spin current dipole in momentum space, analogous to the Berry curvature dipole. It's worth noting that the concept of Berry curvature dipole was thoroughly introduced and its physical significance elucidated in PRL 115, 216806 (2015). Thus, it is surprising that the introduction of this analogous but new concept is confined to just one paragraph. I think the authors should calculate the response up to the second order in the electric field to further clarify the concept.

Additionally, there are several other aspects of the paper that require more in-depth treatment. For instance, the authors briefly mention the Onsager reciprocity relation in line 143. However, Onsager reciprocity in magnetic materials is not straightforward and has been discussed in detail in Physical Review Research 2, 022099R.

Furthermore, the newly added discussion indicates that the interface does not possess any specific symmetry. Consequently, it stands to reason that all components of the SHE and REE tensor should, in principle, be allowed at the interface.

Lastly, the long spin lifetime in MoTe₂ is enigmatic, and the authors correctly point out that it has been reported without a comprehensive explanation. While it may not be feasible to resolve this issue within the scope of the current paper, it would be worthwhile for the authors to highlight the need for further investigation in future research.

In summary, the work presented in this paper is undoubtedly innovative, but it requires a more robust theoretical foundation to merit publication in Nature Communications.

Reviewer #2:

Remarks to the Author:

Generally, I find the authors' response adequate and complete. I can now recommend publication in Nature Communications as is.

Reviewer #3:

Remarks to the Author:

The authors have addressed my concerns.

Response to referees' comments

Reviewer #1 (Remarks to the Author):

The authors have diligently addressed the comments provided by the referees and have made significant revisions to the paper, accompanied by a comprehensive response letter. As highlighted in the previous review, the findings presented in this work are both novel and interesting, making them potentially suitable for publication in Nature Communications. However, I remain concerned about the introduction of the theory presented in Eq. 1 and the subsequent discussion.

Response: We thank the referee for highly appreciating the novelty of our work, and recommending it for publication. We have fully addressed all the questions raised by the referee. The point-by-point responses are listed below.

While the authors have correctly evaluated Eq. 1 based on Wannier parametrization, my primary concern lies with Eq. 1 itself. The authors claim to define a spin current dipole in momentum space, analogous to the Berry curvature dipole. It's worth noting that the concept of Berry curvature dipole was thoroughly introduced and its physical significance elucidated in PRL 115, 216806 (2015). Thus, it is surprising that the introduction of this analogous but new concept is confined to just one paragraph. I think the authors should calculate the response up to the second order in the electric field to further clarify the concept.

Response: We thank the referee for the comment. In Eq. 1, the integrand of the k-space integral is *defined* as the spin current dipole, simply because it corresponds to the k-space dipole moment of the spin current carried by a particular Bloch state. This naming logic is in line with that of Berry curvature dipole in PRL 115, 216806 (2015), as the referee correctly pointed out. *Nevertheless, it should be noted that the emergence of some kind of “dipole” structure in momentum space does not mean it must involve nonlinear response.* Actually, the spin current dipole and Berry curvature dipole appear in very different contexts: the Berry curvature dipole in PRL 115, 216806 (2015) is related to *nonlinear* response effect (second order in E field) for nonlinear charge Hall

conductivity; In contrast, the spin current dipole in the current work is based on *linear* response (first order in E field), *i.e.*, spin Hall conductivity (Eq. 1), and theoretical calculations of first order response to an external electric field are sufficient.

We agree with the referee that due to length restriction, the discussion of Eq. 1 is brief in the main text. In revision, we have added discussion on Eq. 1 in the main text and Methods section. For ease of reviewing, we have provided the revisions in the following, which have been highlighted in red.

“Theoretically, the T-odd SHC can be described by the equation (the derivation is presented in the Methods section)” [Line 15, Page 6].

*“ $D_{ab}^c(\mathbf{k})$ is named as spin current dipole because it corresponds to the k-space dipole moment of the spin current carried by a particular Bloch state. This naming logic is in line with that of Berry curvature dipole. It should be noted that the spin current dipole is related to linear response effect, *i.e.*, spin Hall effect, which is different from Berry curvature dipole related to nonlinear response effect.” [Line 12, Page 13].*

Additionally, there are several other aspects of the paper that require more in-depth treatment. For instance, the authors briefly mention the Onsager reciprocity relation in line 143. However, Onsager reciprocity in magnetic materials is not straightforward and has been discussed in detail in Physical Review Research 2, 022099R.

Response: We thank the referee for bringing this literature [*Physical Review Research 2, 022099R*] to our attention. We have read this reference carefully and found that the conclusion about Onsager reciprocity in magnetic materials in this literature, where the kinetic coefficients are unchanged upon the reversal of time-reversal breaking field (also see *Rev. Mod. Phys.* 82, 1539-1592 (2010)), is fully consistent with our work. In the case of our work, the time-reversal breaking field is the magnetization M of the ferromagnet and the kinetic coefficient is the spin Hall conductivity. To reflect reviewer’s suggestion, we have cited the reference as Ref. 50 in the revised manuscript.

Furthermore, the newly added discussion indicates that the interface does not possess any specific symmetry. Consequently, it stands to reason that all components of the SHE and REE tensor should, in principle, be allowed at the interface.

Response: We thank the referee for this comment. We agree with the referee that all components are allowed from symmetry perspective, as mentioned in our original manuscript. However, the MoTe₂-based non-local spin detector in our FGT/MoTe₂ device is only sensitive to σ_{yx}^z component of *T*-odd charge-to-spin conversion due to the crystalline symmetry of MoTe₂, and thus only the σ_{yx}^z component is required to be considered in the theoretical calculations. We have clarified this point in our manuscript by the statement “*In our measurement configuration, the ISHE in the MoTe₂ is sensitive only to the spin current $J_y^{S_z}$ with spin component out of the transport plane, i.e. spin-z polarized.*” (Page 6, paragraph 1)

Lastly, the long spin lifetime in MoTe₂ is enigmatic, and the authors correctly point out that it has been reported without a comprehensive explanation. While it may not be feasible to resolve this issue within the scope of the current paper, it would be worthwhile for the authors to highlight the need for further investigation in future research.

Response: We thank the referee for the valuable suggestion. According to the reviewer’s suggestion, we have revised the manuscript and highlighted the need for further research on long spin diffusion length in MoTe₂ in the revised manuscript. For ease of reviewing, we have provided the revision in the following, which has been highlighted in red.

“The long spin diffusion length of MoTe₂ can be further verified by our length-dependence transport measurement (Supplementary Note 1). Nevertheless, comprehensive understanding of such a long spin diffusion length observed in MoTe₂ with strong spin-orbit coupling requires more theoretical and experimental effort in the future.” [Line 17, Page 4].

In summary, the work presented in this paper is undoubtedly innovative, but it requires a more robust theoretical foundation to merit publication in Nature Communications.

Response: We thank the referee for highly appreciating the novelty of our work. By fully addressing the comments and suggestions by the referee, we believe that our work is ready for publication in Nature Communications.

Reviewer #2 (Remarks to the Author):

Generally, I find the authors' response adequate and complete. I can now recommend publication in Nature Communications as is.

Response: We thank the referee for recommending the publication of our work in Nature Communications.

Reviewer #3 (Remarks to the Author):

The authors have addressed my concerns.

Response: We thank the referee for recommending the publication of our work in Nature Communications.